# Asymmetry of Gray- and White-Matter Volume and Metabolites in the Central-Vestibular System in Healthy Individuals

**DOI:** 10.3390/jcm12041272

**Published:** 2023-02-06

**Authors:** Ki Jin Kwon, Jae Yong Byun

**Affiliations:** Department of Otorhinolaryngology, Head & Neck Surgery, School of Medicine, Kyung Hee University, Seoul 02447, Republic of Korea

**Keywords:** functional MR spectroscopy, gray-matter volume, brain metabolites, primary vestibular cortex, healthy control, brain asymmetry

## Abstract

This study was designed to determine whether there was an asymmetry of structure and neurochemical activity of the interhemispheric vestibular-cortical system between healthy individuals and patients with vestibular failure. Previous studies have identified differences in gray-matter-volume (GMV) and white-matter-volume (WMV) asymmetry in the central-vestibular system and in concentrations of brain metabolites in the parietal lobe 2 (PO_2_) between patients with vestibulopathy and healthy controls. However, a comparison of the left and right sides in the healthy controls has not been made conclusively. This study included 23 healthy right-handed volunteers, and was carried out between March 2016 and March 2020. A three-dimensional T1-weighted image was used to calculate the GMV and WMV of the central-vestibular network on both sides, and proton magnetic resonance spectroscopy (H1MRS) was employed to analyze the brain metabolites in the PO_2_ area. The relative ratios of N-acetylaspartate (NAA)/tCr, tNAA/tCr, glycerophosphocholine (GPC)/tCr, Glx/tCr, and myo-inositol/tCr were quantified from the proton-MRS data. GMV and WMV differed significantly between the right and left vestibular-cortical regions. The GMVs of the right PO_2_, caudate, insula, and precuneus were significantly higher than those of the same locations on the left side; however, in the Rolandic operculum, the GMV on the left was significantly higher than on the right. In the PO_2_, Rolandic operculum, thalamus, and insula, the WMV on the left side was higher than on the right side of the corresponding location. However, the right caudate and precuneus WMV were higher than the left at the same location. In the H1MRS study, the Glx/tCr and GPC/tCr ratios on the left side were significantly higher than on the right. In comparison, the NAA/tCr and tNAA/tCr ratios showed contrasting results. The NAA/tCr ratio (r = −0.478, *p* = 0.021), tNAA/tCr ratio (r = −0.537, *p* = 0.008), and Glx/tCr ratio (r = −0.514, *p* = 0.012) on the right side showed a significant negative correlation with the participants’ age. There was no relationship between GMV and metabolites on either side. Brain structure and concentrations of brain metabolites related to the vestibular system may differ between the two hemispheres in healthy individuals. Therefore, the asymmetry of the central-vestibular system should be considered when performing imaging.

## 1. Introduction

Unilateral vestibular deafferentation undergoes a recovery process called compensation, but the mechanism is unclear. The vestibular system is classified into peripheral and central parts. The clinical course of patients with vestibular failure is variable, and patients with the same degree of peripheral hyporesponsiveness often exhibit varying functional recovery [1]. This implies that the peripheral vestibular deficit alone does not appear to determine functional recovery. In addition to peripheral regeneration, vestibular compensation may involve central recalibration [2]. There is substantial physiological, clinical, and functional imaging evidence for central vestibular compensation after unilateral peripheral-vestibular lesions [2,3]. Metabolic changes and subsequent adaptive processes occur in response to structural changes, such as changes in brain-tissue volumes (e.g., white-matter volume [WMV] and gray-matter volume [GMV] in the brain). Nevertheless, most studies have been conducted without taking normal controls into account or considering the possibility of asymmetry of brain structure in healthy individuals. A previous study reported volumetric differences between the two hemispheres in healthy individuals [4]. However, there have been no reports of structural brain asymmetry in the central vestibular network. Furthermore, most studies suggest that GMV alteration is related to compensation, and the increase or decrease in GMV and WMV alteration is different for each region of the vestibular cortex [5,6]. This suggests that while gross morphological alterations of the vestibular cortex may be related to functional changes, there are other factors that can cause functional changes in the vestibular cortex. This study aimed to determine the asymmetry of GMV and WMV in the central vestibular network and brain metabolite concentrations in the parietal operculum 2 (PO_2_), which is known to be the human equivalent of macaque area parieto-insular vestibular cortex (PIVC) in both hemispheres of healthy individuals.

## 2. Materials and Methods

### 2.1. Subjects

The study population consisted of 23 healthy volunteers who visited the department of otorhinolaryngology between March 2016 and March 2020. The mean age of the volunteers was 54.2 years, and the standard deviation was 9.5 years (range, 30–69 years). Of the twenty-three healthy volunteers, seven were male, and sixteen were female. Twenty-three healthy volunteers underwent proton magnetic resonance spectroscopy (H1MRS), but the image studies of three participants were lost in the data-delivery process and could not be confirmed; therefore, only twenty participants underwent the imaging study (Figure 1). All volunteers were right-handed and age-matched according to sex. The study was approved by the Research Ethics Committee (IRB 2015-12-025-005) and all patients provided informed written consent.

Exclusion criteria were: (1) history of previous psychiatric, neurological, or neuro-oncological disease; (2) history of administration of ototoxic drugs, stroke, brain trauma, or tumors; (3) abnormalities in brain-magnetic-resonance-imaging (MRI) studies; (4) medications that can affect the brain, viz. antiepileptics, and antipsychotics.

### 2.2. Otological Assessment

All patients were assessed by endoscopic exploration of the tympanic membrane, and history taking was performed to determine eligibility.

### 2.3. Image Acquisition

A sagittal structural three-dimensional (3D) T1-weighted (T1W) image was acquired to evaluate the relationship between blood biomarkers and brain-tissue volume, using a turbo-field-echo sequence with the following parameters: repetition time (TR), 8.1 ms; echo time (TE), 3.7 ms; flip angle (FA), 8°; field-of-view (FOV), 236 × 236 mm^2^, and voxel size, 1 × 1 × 1 mm^3^ (Figure 2). Additionally, T2-weighted turbo-spin-echo and fluid-attenuated inversion-recovery images were acquired to examine brain malformations. MRI images were acquired for each participant using a clinical 3-T MRI system (Ingenia, Philips Medical System, Eindhoven, The Netherlands).

### 2.4. Magnetic-Resonance-Spectroscopy Acquisition

Single-voxel, point-resolved spectroscopy (PRESS) MRS data were acquired with water suppression accomplished with chemical-shift selective-sequence pulses with a bandwidth of 160 Hz, and applied in resonance with the water signal. The scan parameters were TR, 2000 ms; TE, 35 ms; average, 128 ms; spectral bandwidth, 2000 Hz; sampling points, 2048, and scan time, 4.5 min. The region of interest (ROI) was located separately in the right and left posterior parietal operculum, which is known to be equivalent to the PIVC. To localize the voxel location of the spectroscopy, transverse, coronal, and sagittal images were obtained using a 3D T1W imaging sequence. The spectral volume was located on the central portion of the parietal operculum, based on T1 and T2 anatomical images.

### 2.5. Spectral Quantification of MRS Data

The acquired in vivo MRS data were analyzed using a clinical MRS analysis tool supported by the Philips Medical System. The phase was manually adjusted to correct the baseline noise. Metabolite signals were fitted to the chemical-shift range of 4.20 to 0.20 ppm (Figure 3). The metabolites selected to fit the spectrum were as follows: N-acetylaspartate (NAA), N-acetylaspartylglutamate (NAAG), total choline glycophosphocholine, total creatine [tCr = creatine (Cr) + phosphocreatine (PCr)], myo-inositol (mIns), and glutamate and glutamine (Gln) complex (Glx). Finally, we quantified the ratio of the selected metabolites to the total creatine levels.

### 2.6. Image Processing and Analysis

Image processing was performed with Statistical Parametric Mapping version 12 (SPM12) software (http://www.fil.ion.ucl.ac.uk/spm/ (accessed on 1 May 2020)) using the following steps: First, 3D T1W images were fragmented into GMV, WMV, and others, including CSF. Second, the segmented GMV was spatially normalized to the standard template. Finally, the GMV and WMV maps were smoothed with full width at half maximum of 8 × 8 × 8 mm^3^ Gaussian kernels (FWHM). To obtain data on both GMV and WMV from the right and left sides of specific brain areas, we defined the following brain areas using the atlas-based software wfu_pickAtlas (https://www.nitrc.org/projects/wfu_pickatlas/ (accessed on 1 May 2020)): PO_2_, caudate, insula, precuneus, Rolandic operculum, thalamus, and cuneus. Data related to GMV and WMV were extracted with the Marsbar toolbox (http://marsbar.sourceforge.net/ (accessed on 1 May 2020)) from all participants.

### 2.7. Statistical Analysis

All statistical analyses were performed using the SPSS 20.0 software. A paired t-test was used to investigate the asymmetry between the right and left sides of brain-tissue volume and brain metabolites. Pearson’s correlation analysis was performed to investigate the relationships between each MRI measure on each side of the brain and the age. A partial correlation analysis was performed with age as a covariate to investigate the correlation between brain-tissue volume and brain metabolites on each side of the brain. Statistical significance was set at *p* < 0.05.

## 3. Results

### 3.1. Brain-Tissue Volume

Table 1 and Figure 4 list the results of statistical analyses of data relating to GMV and WMV obtained in the specific central-vestibular system. The right-side GMVs were significantly higher than those of the left side in PO_2_, caudate, thalamus, insular, and precuneus. In contrast, the GMV on the left side was higher in the Rolandic operculum. The WMV was significantly higher on the left side than on the right side in the PO_2_, Rolandic operculum, thalamus, and insula, but this was vice versa in the caudate, cuneus, and precuneus.

### 3.2. Brain Metabolites of MRS

The GPC/tCr and Glx/tCr ratios were significantly higher on the left side than on the right. In contrast, the NAA/tCr and tNAA/tCr ratios were significantly higher on the right side than on the left (Table 2 and Figure 5). Upon analysis of the metabolites, the left and right NAA/tCr (Pearson’s correlation, −0.463; *p* = 0.026 and Pearson’s correlation, −0.478; *p* = 0.021, respectively), and right tNAA/tCr (Pearson’s correlation, −0.537; *p* = 0.008) and Glx/tCr (Pearson’s correlation, 0.514; *p* = 0.012) ratios showed significant negative correlations with age (Table 3 and Figure 6). However, no significant correlation was observed between age and GMV (Table 4).

### 3.3. Partial Correlation between GMV and Brain Metabolites

No correlation was detected between the GMV in PO_2_ and brain metabolites on either side (Table 5).

## 4. Discussion

### 4.1. Comparison with Previous Studies

Several studies have used positron emission tomography and functional MRI (fMRI) to study the vestibular cortex area and dominant vestibular cortex to verify the position of the dominant hemisphere according to handedness [1,7]. However, studies on specific brain volumes and brain neurometabolites have not been conducted, to date. Previous studies have confirmed that an increase in GMV was observed in specific areas of the vestibular cortex (i.e., insula, retroinsular vestibular cortex, and superior temporal gyrus) as compensation for vestibular neuritis (VN) [6]. Similar studies also confirmed that gray-matter intensity increased during the VN-associated compensatory process in the superior temporal gyrus [6,8]. The GMV of the caudate, insula, precuneus, and PO_2_, which comprise the vestibular system, was shown to be higher on the right side in normal controls in this study.

As with GMV in the vestibular cortex region, no studies comparing the metabolites in the left and right sides of the brain have confirmed any differences in healthy controls. In a previous study comparing the Glx concentration of PIVC according to low/high visual-attentional load, it was confirmed that the Glx of PIVC decreased when visual attention was strong [9]. In this study, the Glx concentration of the dominant hemisphere was found to be relatively low. Although an experimental study performed in rats showed that choline acetyltransferase activity was increased in the vestibular nucleus complex, no studies have compared the left–right difference of choline in humans [10]. As with Glx, the GPC in PO_2_ was also relatively high on the left side. Although specific studies on the levels of NAA in the vestibular cortex have not yet been confirmed, a report demonstrates that the concentration decreases with aging and neural degeneration in the whole brain [11]. In this study, the concentration of NAA in the right PIVC, which corresponds to the dominant vestibular cortex, was relatively higher than that in the left.

A recent study showed that age did not affect GMV in temporal lobe, auditory and cerebellar networks, confirming that GMV in the vestibular region is independent of age [12]. There is also a study that confirms an age-related reduction in GMV by measuring the PIVC area using diffusion tensor imaging (DTI); however, this study confirmed that there was a reduction in PIVC in the elderly group accompanied by gait impairment and decreased balance ability [13].

A study has reported that serum NAA was lower in the older age group (>60 years) than in the younger age group, and a decrease with age was confirmed in whole-brain NAA using MRS [14,15,16]. In this study, a significant decrease in the NAA/tCr ratio with age was observed in both PIVC areas, which is consistent with the results of a previous study. The concentrations of the mIns/tCr and GPC/tCr ratios did not show differences by direction.

### 4.2. Significance of Vestibular GMV and Brain Metabolites in PIVC Corresponding to Vestibular Cortex

In a previous study, the right-hemispheric dominance was observed in vestibular and eye movements in right-handed individuals, and the opposite result was confirmed in left-handed individuals [1,9]. Research has been conducted in representative regions related to the vestibular region, and differences between the left and right sides have been found in most regions. However, individual interpretation of detailed comparisons of specific areas is difficult because the research area affects various functions, as well as vestibular functions [17]. However, the finding that the GMV of the PIVC area increases after vestibular disorder can be considered to mean that the area plays a major role in compensation. Regarding persistent postural-perceptual dizziness, the finding that GMV decreases in the temporal cortex, cingulate cortex, precentral gyrus, hippocampus, dorsolateral prefrontal cortex, caudate nucleus, and cerebellum suggests that the absolute level of GMV is involved in vestibular function [6,18]. The increase in GMV in the PIVC-like region of the right vestibular cortex, which is the dominant hemisphere in right-handed people, suggests that the right hemisphere plays an important role in vestibular function compared to the left hemisphere in the healthy controls. Structural and functional relationships between unilateral vestibular failure and PIVC were observed in another study on vestibular connectivity using DTI and combined functional MRI [19].

Glu and Gln are components of neurons and glia, and are present in large amounts in the brain. There is an important metabolic interaction in which neurons and glia communicate using Glu and Gln. Glu is released from presynaptic cells and acts on postsynaptic receptors to induce activation [20]. The relative decrease in Glu and Gln of PIVC and the inhibition of PIVC according to visual stimuli have already been addressed in previous studies regarding their correlation [9]. However, in this study, the observation that the Glx of PIVC was relatively low in the non-dominant hemisphere on the left side suggests that the absolute concentration of Glx is not significantly related to vestibular cortex function. Instead, it suggests that the amount of change is significant. Furthermore, this appears to indicate that the absolute value of Glx decreased compared to that in the non-dominant hemisphere, due to the continuous activity of the vestibular function of the dominant hemisphere. Choline is an important nutrient for normal brain development and plays an important role in synthesizing DNA and histone [21]. In this study, like Glx, a relatively high level of choline was observed in the left hemisphere, which is non-dominant. It is difficult to regard choline as a metabolite directly related to vestibular function because choline is highly dependent on lifestyle or diet, and there is no relation to aging, which will be mentioned later [21,22].

NAA is one of the most abundant metabolites in the mammalian CNS, and is known to be involved in neuronal density and mitochondrial function [23]. Several studies using MRS have investigated the increase and decrease in the concentration of NAA and its role in brain metabolism in various diseases. They have shown that NAA levels are strongly associated with disease changes [24]. In a recent study, MRS was performed in twenty patients with vestibulopathy who had normal MRIs, and a decrease in the NAA concentration and NAA/tCr ratio was detected in five patients [25]. Based on the research results showing that there is a correlation between the decrease in the absolute level of NAA and various neuropathies and the fact that NAA is maintained at a relatively high level in the right hemisphere of the PIVC, a core region of the vestibular cortex, it is estimated that the absolute level of the corresponding metabolite is related to the function of the vestibular system.

MIns is naturally produced in the human body. It is the major inositol-containing phospholipid and the primary molecule involved in cellular signal transduction. It plays an important role in osmoregulation, and is used as a glial marker when assessing gliosis and brain metabolic activity [26]. As confirmed by the authors, no studies addressing changes in mIns in the vestibular cortex could be identified. There is a study showing that mIns is associated with neuronal damage or neuropathy, but there was no difference between the two PIVCs in the results of our study. Although further research is needed, the relationship between vestibular function and mIns appears to be relatively weak, and the influence of glial cells in the vestibular area appears rather weak.

As a factor involved in dizziness, a study showed that aging affects the vestibular-ocular reflex (VOR) and vestibulo-spinal reflex, and that the incidence of dizziness increases with aging [27]. However, it has been confirmed, by using a video-head impulse test, that the VOR was maintained continuously before the ages of 80–89, and vestibular compensation was identified as the cause [28]. Therefore, it has recently emerged that cognitive deficits, rather than VOR, cause changes in vestibular function due to aging [29]. According to this study, deterioration of the vestibular cortical network is understood as a functional problem rather than a structural problem of the corresponding structure. In this study, there were no changes in the vestibular cortex GMV according to age in the healthy controls. In contrast, six brain metabolites were investigated; among them, it was confirmed that the NAA/tCr and tNAA/tCr ratios were significantly decreased in variables showing changes with aging.

In our study, Glx showed a significant decrease with age only on the right side. This suggests that the absolute value of Glu and its precursors corresponding to excitatory stimuli decreased with continuous stimulation of vestibular function, as previously hypothesized, and that the decrease in its level in the dominant hemisphere was particularly pronounced. NAA is a marker of neuronal damage. It also plays a role in neuronal metabolism. Many patients with brain diseases showed a decreased NAA/tCr ratio. Changes in NAA levels may occur because of primary neuronal degeneration. In our study, NAA showed a decreasing pattern with age, and its correlation was significant in both directions. Therefore, it would be worthwhile to identify the pattern of Changing value of Glx and NAA in patients with vestibular failure.

MIns is elevated in low-grade astrocytoma, Alzheimer’s disease, and Down’s syndrome, and reduced in old age, glioblastoma, and hepatic encephalopathy glioblastoma [30]. In APP/PS1 transgenic mice that exhibit early Alzheimer’s disease, an increase in Ins was observed in the frontal cortex and hippocampus with aging, but there is no study examining the level of mIns in the vestibular cortex [31]. As this study did not show age-related differences, it is considered that mIns is not closely related to natural aging or changes in vestibular function [31]. In particular, since this experimental group targeted healthy controls under the age of 70 without Hx vestibulopathy, mIns, considered a brain metabolite associated with the disease, appear to be less correlated with simple aging.

Although the correlation between GMV and age was low, a specific metabolite was confirmed to be correlated with age. This may have influenced the difficulty in specifying the region corresponding to the vestibular cortex. Combining these results, it is possible to conclude that, although some information can be derived by examining the role of GMV in the function of the vestibular cortex, more information can be obtained through brain metabolites.

### 4.3. Limitations

Cortical movements during vestibular-nerve stimulation are not symmetrical, and involve the subject’s handedness. Additionally, the direction in which vestibular stimulation is typically applied and the direction of motion of the vestibular nystagmus are known to be influential. This study was conducted only for the right-handed normal group. As there is no study on GMV, WMV, and the metabolites of the left-handed group’s vestibular cortex, the results may seem somewhat insufficient.

Our study had a small sample size, and we did not compare patients with vestibular failure in this study. Follow-up studies are underway, and the results of the present study can be compared to those of the vestibular-disorder patient group in order to compensate for the insufficient statistical values.

## 5. Conclusions

This study provided data on several metabolites and the role of GMV, which are known to correlate. It was confirmed that NAA, Glx, and GPC of the PIVC area show differences between the left and right hemispheres, which will provide important baseline MR-spectroscopy data for future studies of vestibular disorders.

## Figures and Tables

**Figure 1 jcm-12-01272-f001:**
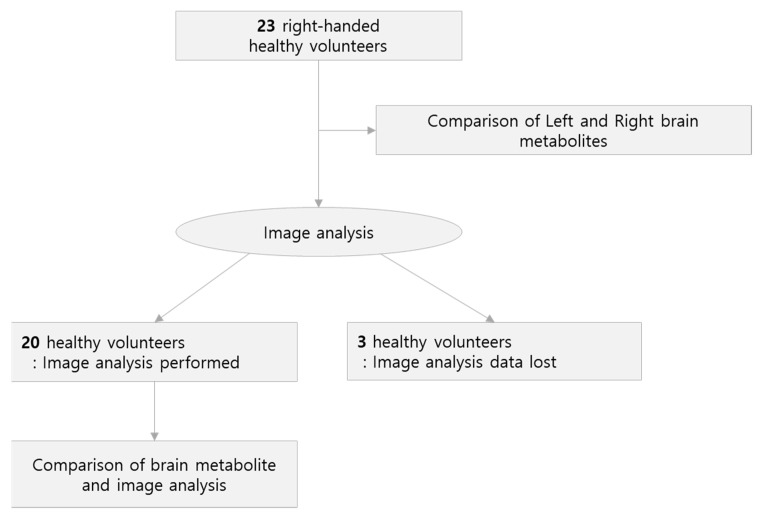
Diagram flow of participant inclusion.

**Figure 2 jcm-12-01272-f002:**
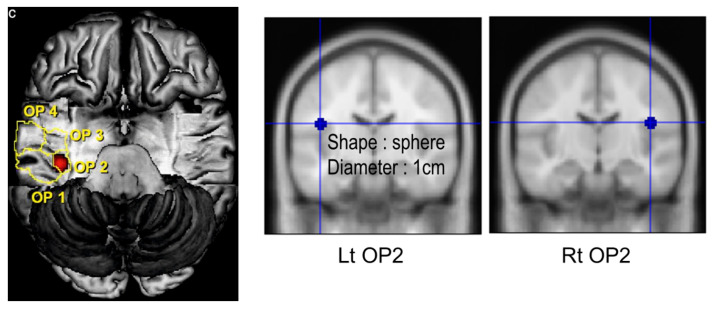
Region of interest for grey-matter volume. Abbreviation: OP, parietal operculum. OP2 is a cytoarchitectonic area located in the posterior parietal operculum, and its volume was measured according to the gray–white-matter ratio. Red and blue dot, Parietal operculum 2.

**Figure 3 jcm-12-01272-f003:**
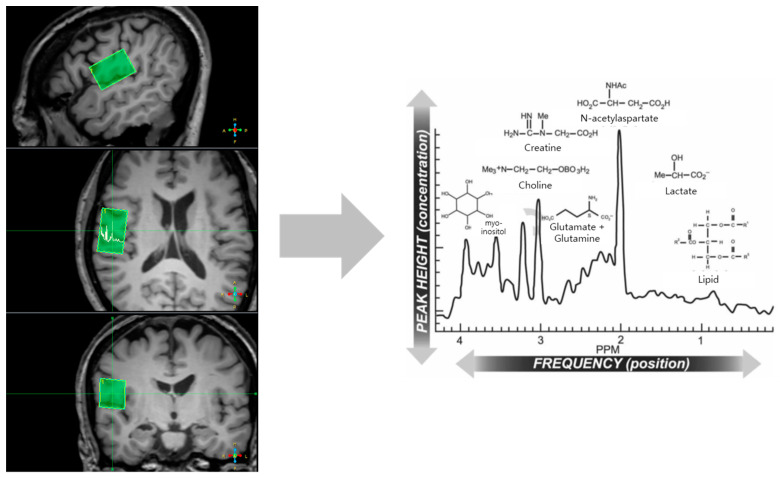
Region of interest for magnetic resonance spectroscopy (MRS). After each metabolite is classified according to ppm through MRS, using the region of interest used to check the GMV and WMV, the concentration was confirmed through the peak height. Abbreviations: PPM, parts per million.

**Figure 4 jcm-12-01272-f004:**
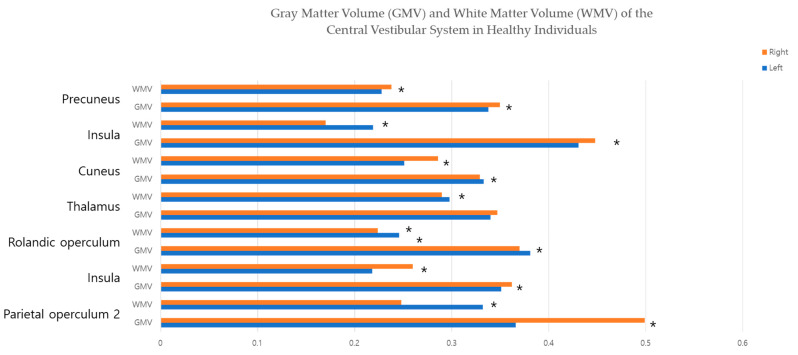
Gray-Matter Volume (GMV) and White-Matter Volume (WMV) of the Central-Vestibular System in Healthy Individuals. * statistically significant.

**Figure 5 jcm-12-01272-f005:**
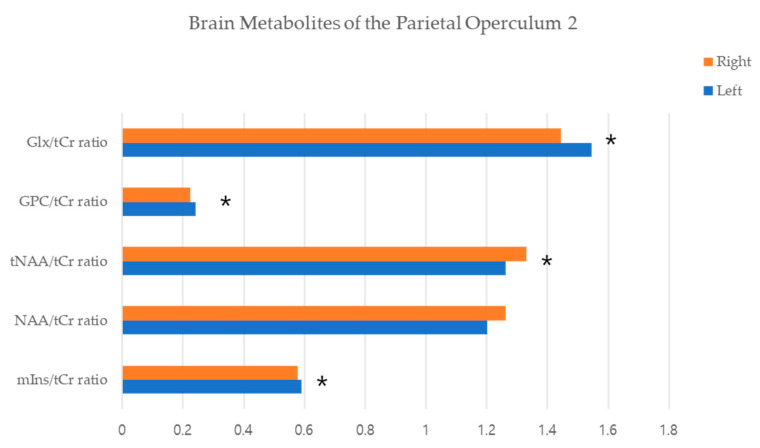
Correlation between metabolite levels and age in the parietal operculum 2. Brain metabolites of the Parietal Operculum 2. Abbreviations: Glx, glutamate and glutamine; tCr, total creatine; GPC, glycophosphocholine; tNAA, total N-acetylaspartate; NAA, N-acetylaspartate; mIns, myo-inositol. * statistically significant.

**Figure 6 jcm-12-01272-f006:**
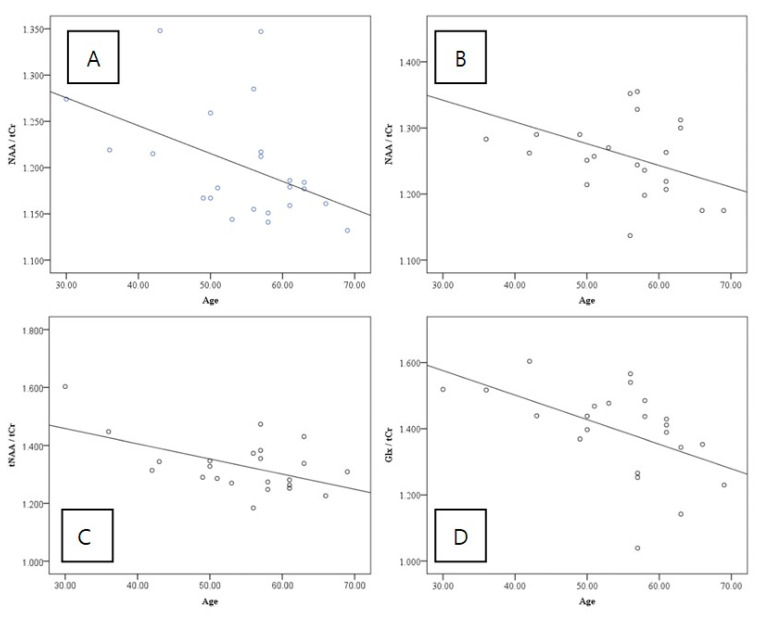
Correlation between metabolite levels and age in the parietal operculum 2. (**A**) Correlation between age and NAA/tCr of left parietal operculum 2. (**B**) Correlation between age and NAA/tCr of the right parietal operculum 2. (**C**) Correlation between age and tNAA/tCr of the right parietal operculum 2. (**D**) Correlation between age and Glx/tCr of the right parietal operculum 2. Abbreviations: NAA, N-acetylaspartate; tNAA, total N-acetylaspartate; tCr, total creatine ratio; Glx, Glutamate and Glutamine.

**Table 1 jcm-12-01272-t001:** Results of Statistical Analyses of Gray-Matter Volume (GMV) and White-Matter Volume (WMV) of the Central-Vestibular System in Healthy Individuals ^a^.

Region of Interest	Left Side (Mean ± SD)	Right Side (Mean ± SD)	Significance (*p*-Value)
Parietal operculum 2			
GMV	0.366 ± 0.071	0.499 ± 0.066	<0.001 *
WMV	0.332 ± 0.034	0.248 ± 0.053	<0.001 *
Caudate			
GMV	0.351 ± 0.041	0.362 ± 0.041	0.001 *
WMV	0.218 ± 0.025	0.260 ± 0.029	<0.001 *
Rolandic operculum			
GMV	0.381 ± 0.051	0.370 ± 0.046	0.038 *
WMV	0.246 ± 0.030	0.224 ± 0.025	<0.001 *
Thalamus			
GMV	0.340 ± 0.048	0.347 ± 0.059	0.266
WMV	0.298 ± 0.041	0.290 ± 0.045	0.020 *
Cuneus			
GMV	0.333 ± 0.048	0.329 ± 0.041	0.543
WMV	0.251 ± 0.038	0.286 ± 0.042	<0.001 *
Insula			
GMV	0.431 ± 0.044	0.448 ± 0.044	0.001 *
WMV	0.219 ± 0.022	0.170 ± 0.016	<0.001 *
Precuneus			
GMV	0.338 ± 0.033	0.350 ± 0.036	0.004 *
WMV	0.228 ± 0.026	0.238 ± 0.028	0.004 *

^a^ 3-T system (repetition time, 8.1 ms; echo time, 3.7 ms; flip angle, 8°; field-of-view, 236 × 236 mm^2^, and voxel size, 1 × 1 × 1 mm^3^; scan time, 5 min) is used to obtain the image. The image is a relative value, analyzed using MATLAB R2018a, SPM 12, and Marsbar toolbox. * statistically significant.

**Table 2 jcm-12-01272-t002:** Brain Metabolites of the Parietal Operculum 2 ^a^.

Metabolite of Parietal Operculum 2	Left Side (Mean ± SD)	Right Side (Mean ± SD)	Significance (*p*-Value)
mIns/tCr ratio	0.591 ± 0.064	0.578 ± 0.049	0.333
NAA/tCr ratio	1.202 ± 0.062	1.262 ± 0.065	<0.001 *
tNAA/tCr ratio	1.263 ± 0.078	1.331 ± 0.092	0.002 *
GPC/tCr ratio	0.242 ± 0.016	0.223 ± 0.039	0.034 *
Glx/tCr ratio	1.545 ± 0.111	1.444 ± 0.026	<0.001 *

Abbreviations: PIVC, parietoinsular vestibular cortex; mIns, myo-inositol; tCr, total creatine; NAA, N-acetylaspartate; tNAA, total N-acetylaspartate; GPC, glycophosphocholine; Glx, glutamate and glutamine. ^a^ This is the value of measuring the peak concentration after confirming the metabolite signal in the range of chemical shift from 4.20 to 0.20 ppm in the region of interest. * statistically significant.

**Table 3 jcm-12-01272-t003:** Correlation Between Metabolite Levels and Age in the Parietal Operculum 2 ^a^.

Metabolite of Parietal Operculum 2	Left Side Pearson Correlation	Left Side *p*-Value	Right Side Pearson Correlation	Right Side *p*-Value
mIns/tCr ratio	0.112	0.610	−0.209	0.339
NAA/tCr ratio	−0.463	0.026 *	−0.478	0.021 *
tNAA/tCr ratio	−0.332	0.121	−0.537	0.008 *
GPC/tCr ratio	0.134	0.542	−0.082	0.723
Glx/tCr ratio	0.134	0.542	−0.514	0.012 *

Abbreviations: mIns, myo-inositol; tCr, total creatine; NAA, N-acetylaspartate; tNAA, total N-acetylaspartate; GPC, glycophosphocholine; Glx, glutamate and glutamine. ^a^ Pearson correlation between age and each metabolite using a normality test. * statistically significant.

**Table 4 jcm-12-01272-t004:** Correlation Between GMV and Age ^a^.

Region of Interest	Left Side Pearson Correlation	Left Side *p*-Value	Right Side Pearson Correlation	Right Side *p*-Value
Parietal operculum 2	0.006	0.980	−0.035	0.884
Caudate	0.064	0.790	0.016	0.947
Rolandic operculum	−0.117	0.623	−0.156	0.511
Thalamus	0.082	0.731	0.049	0.838
Cuneus	0.260	0.269	0.091	0.702
Insula	−0.114	0.632	−0.234	0.322
Precuneus	0.010	0.966	0.142	0.549

^a^ Pearson correlation between age and each region of interest using a normality test.

**Table 5 jcm-12-01272-t005:** Partial Correlation between the GMV of Parietal Operculum 2 and Metabolite ^a^.

Metabolite	Left Side (Partial Correlation)	Left Side *p*-Value	Right Side (Partial Correlation)	Right Side *p*-Value
mIns/tCr ratio	0.138	0.574	−0.059	0.810
NAA/tCr ratio	0.301	0.210	−0.130	0.594
tNAA/tCr ratio	0.329	0.169	0.129	0.598
GPC/tCr ratio	0.003	0.991	0.094	0.701
Glx/tCr ratio	0.280	0.246	−0.062	0.801

Abbreviations: mIns, myo-inositol; tCr, total creatine; NAA, N-acetylaspartate; tNAA, total N-acetylaspartate; GPC, glycophosphocholine; Glx, glutamate and glutamine. ^a^ After normality test, right parietal operculum 2 and right metabolite, left parietal operculum 2 and left metabolite were compared. This is the result of each partial correlation, with age as a covariate.

## Data Availability

Not applicable.

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
