# Peer review of "Asymmetry of Gray- and White-Matter Volume and Metabolites in the Central-Vestibular System in Healthy Individuals"

_jcm, 2023, doi:10.3390/jcm12041272_

Round 1
Reviewer 1 Report
The manuscript investigates the interhemispheric asymmetry of structure and neurochemicals in the central vestibular network of healthy individuals. However, the manuscript is difficult to read because of the way it is written. I would strongly recommend to improve the writing and grammar and make a better flow of sentences which is easier to read. For example, line 11 the term “To date…” was not clear to me in this sentence.
It is also not clear to me what authors are trying to say in line number 177-179. To me it appears like a careless editing.
It will be easier to follow the results if the asymmetry is presented in the form of laterality index.
In line number 51-53, are the authors refuting the claim of increased GMV during compensation?
Out of 20 subjects, how many were females? was there any difference between male and females asymmetries?
I can't understand what authors are trying to say in line 323-325. Are they saying that the inference can change as the conc can change in the the future study OR it is the conc change that matters rather than the absolute value. Please reframe the sentence
Author Response
Thank you for considering this manuscript for publication in your journal. The comments from the editor and the reviewers were very helpful. Herein are our responses to those comments. We have also revised the manuscript as suggested by the editor and reviewers. We tried as much as possible to retain the content of the original article during the revision.
We hope this revised manuscript will be suitable for publication in “JCM”
Extensive editing of English language and style was performed using the following Editage (https://app.editage.co.kr/orders/download-files/KHNMC_560_2). Since the Editing Certificate cannot be attached separately, only the corresponding link is presented.
Thank you.

Reviewer 2 Report
The manuscript entitled “Regional Gray, White Matter Volume, and Metabolites of Interhemispheric Asymmetry in the Central Vestibular System of Healthy Individuals” reported on an interesting topic regarding as an asymmetry of structure and neurochemical activity of the interhemispheric vestibular cortical system in 23 healthy individuals in order to examine the presence of an asymmetry of structure and neurochemical activity of the interhemispheric vestibular cortical system. The author used a 3-dimensional T1-weighted image to calculate GMV and WMV of the central vestibular network on both sides and proton MR spectroscopy (H1MRS) to collect brain metabolites in the PO2 area. They found that GMV and WMV differed significantly between the right and left vestibular cortical regions.
The text and contents are understandable.
Only some Specific concerns:
Of the 23 healthy volunteers, seven were male and 16 were female. Was there any differences linked to the sex?
In the exlusion criteria please add information about the use of other drugs other than ototoxic drugs, like antipileptic medication
In my opinion Hight density EEG could be useful to confirm these findings in an increased number of subjects.
Author Response
Thank you for considering this manuscript for publication in your journal. The comments from the editor and the reviewers were very helpful. Herein are our responses to those comments. We have also revised the manuscript as suggested by the editor and reviewers. We tried as much as possible to retain the content of the original article during the revision.
We hope this revised manuscript will be suitable for publication in “JCM”

Reviewer 3 Report
The aim of this this study was to determine whether there was an asymmetry of structure and neurochemical activity of the interhemispheric vestibular cortical system in healthy individuals compared to patients with vestibular failure.
Line 53-58 - This study aimed to determine the asymmetry of GMV and WMV in the central vestibular network and brain metabolite concentrations in the parietal operculum 2 (PO2), which is known to be the human equivalent of macaque area parieto-insular vestibular cortex (PIVC) in both hemispheres of healthy individuals who will be compared to patients with vestibular failure, which will be treated in a future study.
In the abstract section authors stated that the aim of the study was to compare healthy individuals to patients with vestibular failure while in introduction section there is an information that patient will be treated in a future study. So no patients were included in the study and analysed.
Authors did not provide sufficient background in the introduction section. The section could be more explained in details.
The materials and methods section is adequately described.
Discussion section
Line 187: Several studies have verified the position of the dominant hemisphere according to 187 handedness using positron emission tomography and functional MRI (fMRI) in the ves- 188 tibular cortex area and dominant vestibular cortex.1 On what study results do authors indicate that (reference is missing)?
Also line 190: Previous studies 190 confirmed that an increase in GMV was observed in specific areas of the vestibular cortex 191 (i.e., insula, retroinsular vestibular cortex, and superior temporal gyrus) as compensation 192 for vestibular neuritis (VN). No references
Line 211: A recent study showed that age did not affect GMV in the temporal, auditory, and 21
cerebellar networks. Which study? Author often use expression: this study (meaning authors study??), another study, it is hard to follow the idea
Also line 274: As confirmed by the authors, studies on changes in the concentration of mIns limited to the vestibular cortex have not been reported, and an overall increase or decrease in mIns in the brain has been reported in various diseases, including neuronal damage and neuropathy. What was confirmed by the authors? Did the authors do another research on various desieses in that topic or is it another study – lack of reference.
Line 271: MIns are naturally produced in the human body. It is the major inositol-containing
… so are or is?. Please be consistent.
Glu, NAA, MIns definitions are rather suitable for introduction section.
Line 292: This suggests that the absolute value of Glu and its precursors corresponding to excitatory stimuli decreased with continuous stimulation of vestibular function, as previously hypothesized, and that the decrease in the dominant hemisphere was particularly pronounced. Where are the hypothesis? They were not included in the introduction section.
Line 304: In APP/PS1 transgenic mice that induce early Alzheimer’s disease, an increase in Ins was observed in the frontal cortex and hippocampus with aging… lack of reference.
Line 320: This study has limitations in that other factors involved were excluded, as the experiment was conducted only in the right-handed normal group. The sentence makes no sense.
Discussion section is rather a continuation of introduction and results section rather than actual discussion.
Conclusions section:
The conclusions are inconsistent with the aim of the study.
Line 328: To analyze the characteristic features of patients with vestibular failure from MR spectrography data from previous studies, baseline data from healthy controls should be acquired. Therefore, this study is essential to establish baseline MR spectrography data. – while the authors aimed to determine whether there was an asymmetry of structure and neurochemical activity of the interhemispheric vestibular cortical system in healthy individuals compared to patients with vestibular failure. Plus Patients were not included in the study
Author Response

(The authors gave the same response as above.)

Round 2
Reviewer 3 Report
Thank you fore the revision and corrections of the manuscript